# Different Pathogen Defense Strategies in *Arabidopsis*: More than Pathogen Recognition

**DOI:** 10.3390/cells7120252

**Published:** 2018-12-07

**Authors:** Wei Zhang, Feng Zhao, Lihui Jiang, Cun Chen, Lintao Wu, Zhibin Liu

**Affiliations:** 1College of Bioengineering, Sichuan University of Science and Engineering, Zigong 643000, China; zhangwei19840117@163.com (W.Z.); jianglihui19940625@163.com (L.J.); 2Helmholtz Zentrum München, Department of Environmental Sciences, Institute of Biochemical Plant Pathology, Ingolstädter Landstr. 1, 85764 Neuherberg, Germany; feng.zhao@helmholtz-muenchen.de; 3College of Chemistry and Life Science, Chengdu Normal University, Chengdu 611130, China; chencun0211@126.com; 4West Rape Research Institute, Guizhou Academy of Agricultural Sciences, Guizhou 550008, China; wult12@126.com; 5Key Laboratory of Bio-Resources and Eco-Environment of Ministry of Education, College of Life Sciences, Sichuan University, Chengdu 610064, China

**Keywords:** pathogen defense, PTI, ETI, hormone pathways, small molecules, SAR, priming

## Abstract

Plants constantly suffer from simultaneous infection by multiple pathogens, which can be divided into biotrophic, hemibiotrophic, and necrotrophic pathogens, according to their lifestyles. Many studies have contributed to improving our knowledge of how plants can defend against pathogens, involving different layers of defense mechanisms. In this sense, the review discusses: (1) the functions of PAMP (pathogen-associated molecular pattern)-triggered immunity (PTI) and effector-triggered immunity (ETI), (2) evidence highlighting the functions of salicylic acid (SA) and jasmonic acid (JA)/ethylene (ET)-mediated signaling pathways downstream of PTI and ETI, and (3) other defense aspects, including many novel small molecules that are involved in defense and phenomena, including systemic acquired resistance (SAR) and priming. In particular, we mainly focus on SA and (JA)/ET-mediated signaling pathways. Interactions among them, including synergistic effects and antagonistic effects, are intensively explored. This might be critical to understanding dynamic disease regulation.

## 1. Introduction

Since they exposed to challenging environments, plants face great threats from pathogen infections. Pathogens can be classified into three categories—biotrophic, necrotrophic, and hemibiotrophic pathogens—according to their different infection strategies [1]. Biotrophic pathogens first infect the epidermal cells, and then develop haustoria to contact the plant cells to uptake the nutrients from the living cells. Most of the biotrophic pathogens are host-specific, including *Phytophthora parasitica*, *Erysiphe orontii*, and *Erysiphe cichoracearum* [1]. Necrotrophic pathogens kill plant cells first, and then feed on the dead tissues. Most of the necrotrophs infect a broad range of hosts, except for a few ones. The fungal pathogens *Botrytis cinerea*, *Alternaria brassicicola*, and *Sclerotinia sclerotiorum* are taken as general examples of necrotrophic infections that kill hosts using toxic metabolites, enzymes, or microRNAs [2,3,4]. For instance, *B.cinerea* small RNAs hijack the host RNA interference machinery to achieve virulence [5]. Hemibiotrophic pathogens will first experience biotrophic lifestyle and then enter a necrotrophic mode. The bacterial pathogen *Pseudomonas syringae* is often taken as a hemibiotroph [6].

During the long-term battle with pathogens, plants have developed sophisticated defense mechanisms to fight pathogen attacks. [1,7,8]. Here, we will summarize and explore recent findings on pathogen-associated molecular pattern (PAMP)-triggered immunity (PTI) and effector-triggered immunity (ETI), especially the roles of the three classical hormone pathways: salicylic acid (SA) and jasmonic acid (JA)/ethylene (ET) pathways (Figures 1 and 2), as well as the cross-talk among them (Figure 3), and other defense mechanisms. First, we provide an overview of PTI and ETI briefly, since many studies have already intensively discussed them [7,9,10]. Second, most of what we discuss falls into three pathways. We provide a very detailed review of the dynamic regulations of SA-JA/ET cross-talk. Finally, we explore the roles of newly identified small molecules in defense, and other interesting defense aspects, including systemic acquired resistance (SAR) and priming. The regulation by hormone pathways involved in small molecules-mediated defenses such as pipecolic acid, SAR, and priming are also discussed. Most of what we discuss concerns research performed in *Arabidopsis*; however, examples from other species are introduced as well, where appropriate.

## 2. Different Layers of Plant Defense

Plants are sessile, and thus, they have developed effective methods to fight against pathogens. Once pathogens have overcome physical barriers such as the cell wall, the most important issue for defense will be the pathogen’s perception. A “zigzag” model of plant immunity was given by Jones and Dangl to describe the evolutionary development of the plant immune system [7]. Similar to animals, plants have evolved an innate defense response that is able to recognize conserved PAMPs via pattern recognition receptors (PRRs) on the plant cell surface. Phase 1 is PAMP-triggered immunity (PTI) as a classical basal resistance is the phase 1. In phase 2, pathogens release effectors to interfere with PTI. In phase 3, single effectors are perceived by nucleotide-binding leucine-rich receptors (NB-LRRs). This recognition triggers ETI, resulting in a much stronger disease resistance. In phase 4, pathogens develop new effectors to suppress ETI [7].

### 2.1. Physical and Chemical Defenses

Due to the constant exposure to a broad range of pathogens, plants have evolved complicated immune systems, including different aspects of defense mechanisms, to protect against pathogen colonization [1,7,11,12,13,14]. Firstly, the primary mechanical and structural barriers prevent the physical penetration of pathogens. The plant cuticle coating the surface of epidermal cells is a protective layer against biotic stresses, and it consists of lipids, hydrocarbon polymers, and waxes [15]. Stomata localized on the leaf surface are natural openings for pathogen entry, and plant cells can close them when challenged by pathogens, such as for instance, *Pseudomonas syringae* [16]. Lenticels are also natural pores that are easily invaded by pathogens. For instance, the area per fruit surface of lenticels positively correlates with plant susceptibility toward *Botryosphaeria dothidea,* which is an apple disease [17]. The importance of the cell wall as a mechanical barrier was elaborated well by the enhanced susceptibility toward pathogens of plants impaired in defense-induced lignification, which can strengthen the cell wall [18]. Recently, defense-induced lignification was found to be regulated by the *Arabidopsis* SG2-type R2R3-MYB transcription factor, MYB15 [19]. Secondly, for chemical defense, plants produce non-specific secondary antimicrobial chemicals to counteract pathogen invasion. Phytoalexins, acting as antimicrobial toxins, can be synthesized de novo very rapidly in response to pathogen infection [20]. The synthesis of camalexin, as a major phytoalexin in *Arabidopsis*, can be regulated by two major enzymes: *Arabidopsis* cytochrome P450 monooxygenase 71A13 (CYP71A13), and *Arabidopsis* cytochrome P450 monooxygenase 71B15/ phytoalexin-deficient 3 (PAD3). The increased susceptibility of the mutant *pad3* toward *Alternaria brassicicola* was caused by camalexin biosynthesis impairment [21]. Phytoanticipins are constitutively produced metabolites against pathogens in plants [22]. Glucosinolates, as well-characterized phytoanticipins, play a pivotal role in defense against herbivores, bacteria, and fungi [23,24]. Glucosinolates are produced in the trichomes of *Arabidopsis*, and their biosynthesis is regulated by the basic helix–loop–helix (bHLH) transcription factors (TFs): MYC2, MYC3, and MYC4 [25]. Plant defensins are small cysteine-rich peptides that are active against bacteria and fungi [26,27,28]. The importance of defensins in killing pathogens are well-proven in many plant species, including *Arabidopsis* [29], *Solanum lycopersicum* (tomato) [30], and *Nicotiana tabacum* [31].

Physical and chemical defenses play significant roles in defending pathogens. However, in the future, the signals from pathogens that direct these responses should receive more research attention.

### 2.2. Basal Disease Resistance Regulated by Pattern Recognition Receptors

PTI takes the main role in the combat against pathogens as a basal resistance for susceptible hosts, and in non-host resistance [7]. PAMPs comprise conserved structure components, such as flagellin and fungal chitin, as well as conserved elongation factor, from pathogens, for instance, elongation factor Tu (EF-Tu) [32,33]. Flagellin 22 (flg22), a 22-amino-acid peptide, is detected by flagellin-sensing 2 (FLS2), encoding a leucine-rich repeat receptor kinase (LRR-RK) [34]. The alteration of FLS2 conformation, due to its interaction with flg22, causes an association with brassinosteroid insensitive 1(BRI1)-associated receptor kinase 1 (BAK1), thus transducing the signal [35]. Furthermore, β-aminobutyric acid (BABA)-responsive L-type lectin receptor kinase-VI.2 (LecRK-VI.2) is necessary for flg22-induced PTI, through an unknown mechanism [36]. Chitin produced by the fungal cell wall is detected by chitin elicitor receptor kinase 1 (CERK1), which is another receptor-like kinase that is able to regulate pathogen defense [37]. Upon chitin recognition, CERK1 can associate with, and phosphorylate, the receptor-like cytoplasmic kinase (RLCK) family protein, avrpphb susceptible 1 (PBS1)-like kinases (PBL27). PBL27 further interacts with and phosphorylates mitogen-activated protein kinase kinase kinase 5 (MAPKKK 5), thus activating the downstream MAPK cascade in *Arabidopsis* [38]. Similarly, OsCERK1 phosphorylates OsRLCK185, which is an ortholog of PBL27 in rice. Then, OsRLCK185 phosphorylates OsMAPKKK 11 and OsMAPKKK 18, which are rice orthologs of MAPKKK 5, transducing the downstream MAPK signaling [39,40]. EF-Tu is recognized by the elongation factor-Tu receptor (EFR), which is another LRR-RK [41]. EFR physically interacts with BAK1, thereby rendering a transduction for defense signaling [42,43]. Endogenous elicitors produced from damaged plant cell fragments from pathogens are characterized as damage-associated molecular patterns (DAMPs). The recognition of DAMPs resembles the recognition of PAMPs. These endogenous elicitors, including cell wall fragments, are released by microbial hydrolysis, cutin monomers, and short peptides, such as systemin and hydroxyproline-containing glycopeptides (HypSys), as well as the 23-amino-acid peptide 1 (PEP1) [44,45,46]. Similar to the perception of flg22, PEP1 is recognized by PEP1 receptor 1 (PEPR1) and PEP1 receptor 2 (PEPR2) [44]. Botrytis-induced kinase 1 (BIK1), belonging to the receptor-like cytoplasmic kinase (RLCK) family, directly interacts with, and is phosphorylated by PEPR1, and likely PEPR2, upon ethylene (ET) treatment. The positive role of PEPR1, PEPR2, and BIK1 in mediating ethylene (ET)-induced defense signaling was proven by Jian-Ming Zhou et al. [47]. In addition, another study indicated that PEPRs and the ET pathway play an additive role in PTI in general [48]. Interestingly, the above-mentioned FLS2, EFR, and PEPRs can be internalized into endosomes requiring BAK1. Endocytosis participates in regulating FLS2-mediated immunity, including stomatal closure and callose deposition [49].

In *Arabidopsis*, more than 600 RLKs exist [50], forming one of the biggest protein families. Most of them have the characteristics of an extracellular domain, a transmembrane domain, and a kinase domain facing the cytoplasm [50]. They play diverse roles in plant activities, including defense [51]. The above-mentioned pattern receptors: FLS2, LecRK-VI.2, EFR, CERK1, and PERPs all belong to receptor-like kinases (RLKs), suggesting an important role of RLKs in mediating defense via binding to specific ligands [52]. FLS3, another RLK, can recognize the epitope of flagellin, and it is termed flgII-28 in *Solanum pimpinellifolium* (tomato) [53]. In *Oryza sativa* (rice), kinase XA21 mediates the resistance of *Xanthomonas* [54]. Receptor-like proteins (RLPs) lacking the kinase motif, and thus different from RLKs, have been implicated in the involvement of defense in *Solanum pimpinellifolium* (tomato)*, Gossypium barbadense* (cotton), and *Brassica napus* [55,56,57]. Fifty-seven putative RLPs were reported to exist in *Arabidopsis* [58]; RLP23 specifically recognizes nlp20, which is a PAMP from necrosis and ethylene-inducing peptide1-like proteins (NLPs) [59]. The RLP protein Ve1 mediates the recognition of *Fusarium oxysporum* in tomato [60]. Recently, the transcriptional regulation of receptor-like protein genes has been systemically analyzed after different stresses and hormone treatments in *Arabidopsis* [61]. *RLP20*, *RLP21*, *RLP22,* and *RLP40* are responsive to *Pseudomonas syringae* infection [62]. However, whether these RLPs play roles in recognizing pathogen epitopes requires more research.

### 2.3. Virulent Effector-Induced Plant Defense

During the long-term battle between pathogens and plants, pathogens have evolved a mechanism involving a type III secretion system that is used to inject effectors to interfere with an unspecific PTI, in order to achieve a successful infection. Many virulent effectors are known to suppress PTI [62,63]. It has been shown that a *Pseudomonas syringae* type III effector can suppress PTI by the downregulation of the putative secreted cell wall and defense proteins [64]. Avirulent Pseudomonas syringae pv tomato strain DC3000-specific B (AvrptoB), as an E3 ubiquitin ligase from *Pseudomonas syringae*, can target the FLS2 receptor for degradation in *Arabidopsis* and mitogen-activated protein kinase 2 in *N. benthamiana* [65]. In addition, hypersensitive response and pathogenicity (Hrp) outer protein (hop) M 1 (HopM1), a conserved *P. syringae* virulence protein, was found to interact with *Arabidopsis* HopM interactor 7 (AtMIN7), which is an immunity-related vesicle traffic regulator, and interfere with its stability via proteasome degradation to suppress PTI [66]. Recently, HopM1 triggers an oxidative burst and stomatal immunity in an AtMIN7-independent manner; however, it most likely acts on AtMIN10 [67]. Generally, the virulent effectors function as the eukaryotic enzymes, such as ubiquitin ligases, phosphatases, and proteases, to regulate immune signaling [68,69,70]. To overcome the virulence of effectors from pathogens, plants, in turn, establish ETI by recognizing virulent effectors via corresponding receptor resistance (R) proteins in a specific gene-to-gene manner, which is commonly accompanied by a hypersensitive response in the form of rapid cell death, which limits the spread of the pathogen from infection sites. Increased SA accumulation is accompanied by the hypersensitive response in ETI [71,72]. The specific recognition of variable virulent effectors is regulated by many R proteins. The effectors recognized by R proteins become nontoxic for the hosts, since R proteins can detoxify the virulence of the effector s. Thus, an avirulent pathogen is a pathogen carrying an effector, which can be recognized by R proteins in plants. The receptor proteins mainly contain nucleotide-binding (NB) domains and LRRs, which can mainly be classified into two subclasses: Toll-interleukin-1 receptor-nucleotide binding-leucine-rich repeat (TIR-NB-LRR) proteins and coiled-coil (CC)-NB-LRR proteins. Perception by NB-LRR proteins can be mediated either through direct interaction with effectors, or by the accessory proteins, whose conformations can be altered by their associations with effectors. Then, the activation of defense signaling is conducted by the alteration of NB-LRR proteins conformations [73]. However, R protein-regulated ETI mostly exists in biotrophic pathogens, and it seldom exists in necrotrophic pathogens. Only one example of R protein-regulated ETI, in that the protein Toll/interleukin 1 receptor triggered a broad defense against several necrotrophs, was reported [74]. Conversely, a specific protein was often found to trigger the disease susceptibility in necrotrophs, which was designated as effector-triggered susceptibility [75].

Future research studies will reveal more novel strategies that are employed by pathogen effectors, which can be exploited to control plant disease.

### 2.4. The Interplay of PTI and ETI

Although PTI and ETI recognize different pathogen patterns, generally, they trigger quite similar responses with mostly overlapping genes expression that suggests their synergistic effects in PTI and ETI [76]. The non-expressor of PR genes 1 (NPR1) is the master regulator in the SA pathway, and is required for the full activation of both PTI and ETI, and especially the cell death as triggered by ETI. Some transcriptional factors, including WRKY DNA-binding proteins (WRKYs) and Translocon-binding transcription factor 1 (TBF1), play pivotal roles in PTI and ETI. TBF1 binds to the *cis* element (GAAGAAGAA) in the promoter regions to upregulate defense genes [77]. Enhanced disease susceptibility 1 (EDS1) can regulate SA synthesis during PTI to suppress pathogen growth [78]. Furthermore, EDS1 physically interacts with the resistances of three TIR-NB-LRR proteins resistance to *Pseudomonas syringae* 4 and 6 (RPS4 and RPS6). Interestingly, EDS1 was found to interact with the bacterial effector AvrRPS4 and HopA1 [76,79]. Thus, EDS1 is also the crucial regulator for both PTI and ETI.

## 3. Hormones in Plant Defense

### 3.1. The SA Pathway in Plant Defense

Downstream of PTI and ETI, the crucial role of signaling pathways such as the SA pathway, JA pathway, and ET pathway in pathogen defense responses is well-established [12]. Biotrophic pathogens induce the SA pathway, and necrotrophic pathogens upregulate both the JA and ET pathways [12]. The precursor chorismate can be converted to SA via two different enzymatic pathways. One is metabolized by phenylalanine ammonia lyase (PAL), while the other involves isochorismate synthase 1/salicylic acid induction deficient 2 (ICS1/SID2). SID2 is mainly responsible for the stress-induced SA accumulation as in *sid2* mutants; the level of SA is only 5–10% of control values after infection or UV stress [80,81]. Most of the SA is converted to SA O-β-glucoside (SAG) by SA glucosyltransferases, and subsequently transported to vacuoles for storage. Small amounts of SA are converted by conjugation to form a salicyloyl glucose ester (SGE) and methyl salicylate (MeSA). MeSA is an inactive form, but it plays a role as a long-distance signal in transferring SAR in tobacco and *Arabidopsis* [71,82]. EDS1 and phytoalexin-deficient 4 (PAD4) are involved in upregulating SA synthesis [71]. NPR1 is a key regulator of SA-mediated responses [83]. In non-challenged cells, NPR1 exists in the cytoplasm of the cell as oligomers linked via intermolecular disulfide bonds. Upon stress, SA-triggered cellular redox change induces the monomerization of NPR1 via the thioredoxins (TRX)-H3 andTRX-H5 [84] (Figure 1). A gene encoding a serine/threonine (S/T) kinase SNF1-related protein kinase 2.8 (SnRK2.8) interacts with and phosphorylates NPR1. The phosphorylation of NPR1 by SnRK2.8 is not triggered by SA; however, it is required for NPR1 nuclear localization to fulfill SA-dependent defense responses. The phosphorylation of NPR1 at serine 589 is indispensable for NPR1 nuclear localization [85] (Figure 1). NPR1 is transported into the nucleus through the nuclear pores [86]. After the importing of monomers of NPR1 to the nucleus, NPR1 interacts with a class of positive TGA transcriptional factors to positively regulate the SA-responsive genes such as pathogen-related proteins [14]. These pathogen-related (PR) genes encode many antimicrobial proteins [87]. For instance, some can destroy the integrity of the pathogen cell wall as lytic enzymes [87]. Many regulatory proteins in the SA pathway can be found to interact with NPR1, such as the NPR1-interacting proteins non-inducible immunity 1 (NIM1)-interacting 1 (NIMIN1), 2, and 3, and Suppressor of *npr1-*inducible 1 (SNI1). These NIMINs play roles in inhibiting the promoter activity of defense genes, which is likely via influencing TGACG sequence-specific binding protein (TGA) transcriptional factors. The binding partner and regulation mechanism of SNI1 by far is still unclear. The *sni1* mutant was screened to rescue the SA response in *npr1* mutants, suggesting an NPR1-independent response. In addition, the constitutively enhanced SA response in suppressor of SA insensitivity 2 (*ssi2*) was confirmed to be partially dependent on SA, but not on NPR1. Moreover, the transcriptional factor glutaredoxin C9 (GRXC9) is another example that regulates a SA-dependent, although NPR1-independent, defense response [88]. The NPR1-independent pathway may require the whirly (WHY) transcription factor family (Figure 1). Probably, NPR1 regulates the downstream SA response, in concert with WHY transcription factors [71]. The synthetic elicitor 3,5-dichloroanthranilic acid (DCA) can induce both NPR1-denpendent and NPR1-independent mechanisms of disease resistance in Arabidopsis. In the *npr1* mutant, DCA-induced resistance was only mildly affected [89]. This was in contrast to the complete loss of pathogen resistance activation by SA, 2,6-dichloro-isonicotinic acid (INA), and benzo(1,2,3)thiadiazole-7-carbothioic acid S-methyl ester (BTH) in *npr1* [90,91]. Moreover, *PR1* could be further induced in the *npr1* mutant by the infection of different accessions of *Botrytis cinerea* indicating other unknown signals bypassing NPR1 [92]. The NPR1 paralogues NPR3 and NPR4 were reported to bind SA as SA receptors; however, NPR1 cannot bind SA, and thus is not an SA receptor. NPR3 and NPR4 work as ubiquitin activating enzyme 3 (E3) ligases to degrade NPR1. The model suggests that high levels of SA promote the interaction of NPR3 and NPR1 to degrade NPR1 to prevent an over-accumulation of NPR1, while low levels of SA inhibit the association of NPR4 and NPR1 to keep basal NPR1 level [93]. However, other studies showed that NPR1 could bind SA with high affinity [94,95]. Instead of being E3 ligases, NPR3 and NPR4 function as transcriptional repressors to interact with TGAs to regulate downstream defense genes such as *WRKY70,* since TGAs could play a negative role in mediating the defense response [95] (Figure 1).

Many loss-of-function mutants leading to retarded SA response and accompanied by a reduced resistance to biotrophic pathogens were thus identified to positively regulate the SA pathway. For instance, genes in involved in kinase-activated signaling such as enhanced disease resistance 1 (EDR1), mitogen-activated protein kinase 3 (MPK3), and MPK6, probably act upstream of SA synthesis [96,97,98,99,100]. Nucleoside diphosphate linked X 6 (NUDT6) functioning in changing reduced nicotinamide-adenine dinucleotide (NADH) metabolism was reported to positively regulate the SA pathway, dependent on NPR1 [101]. In contrast, many mutants constitutively activating the SA response have been found displaying elevated SA accumulation and expression of the *PR* genes. These well-characterized mutants include the *mitogen-activated protein kinase (mpk4*)*, accelerated cell death (acd), lesions simulating disease (lsd), constitutive expression of PR genes (cpr), mildew resistance locus o (mlo), hypersensitive response-like lesions 1 (hrl1), HR-like lesion mimic (hlm1), suppressor of NPR1-1 (sni1) and suppressor of salicylic acid insensitivity (ssi1)* mutants [102,103,104,105,106,107,108,109,110]. A group of WRKY transcription factor genes such as WRKY70, WRKY33, and WRKY18 were found to modulate the SA pathway via genetic analysis. WRKY70, WRKY50, and WRKY51 positively regulate the SA pathway, whereas WRKY33 exerts a negative influence on SA pathways [111,112,113,114]. Many *WRKY* transcription factors are induced after the nuclear translocation of NPR1 monomers [115].

### 3.2. The JA Pathway in Plant Defense

The JA pathway mainly protects plants against necrotrophic pathogens and wounding. The plant hormone JA is a lipid-derived compound, and it is synthesized through the oxylipin biosynthetic pathway. The JA synthesis starts with the precursor α-linolenic acid, which is released from membrane lipids in chloroplast. The key synthetic enzymes include a 13-lipoxygenase (13-LOX), a 13-allene oxide synthase (13-AOS), an allene oxide cyclase (AOC), and an OPDA-specific reductase (OPR3). There are many conjugation forms of JA, including jasmonoyl-l-isoleucine (JA-Ile), methyl jasmonate (MeJA), jasmonoyl -1-aminocyclopropane-1-carboxylic acid (JA-ACC), jasmonoyl-1-*O*-β-glucose (JA-Glc), and 12-hydroxyjasmonic acid sulphate (12-HSO4-JA). However, there is evidence showing that only MeJA and JA-Ile are the active forms [116]. Upon stress, JA can be rapidly conjugated to amino acids such as isoleucine by jasmonate-resistant 1 (JAR1), leading to the major biologically active form JA-Ile. JA-Ile can then bind to the F-box protein coronatine insensitive 1 (COI1), leading to the conformation change of COI1. These conformation changes allow for the association of COI1 with the jasmonate zinc-finger expressed in inflorescence meristem (ZIM) (JAZ)-domain transcriptional repressor proteins. Inositolpentakis phosphate 5 (InsP5) acts as a co-receptor for JA-Ile to stabilize the association of JAZ-COI1 [117]. COI1 functions in the E3-ligase S phasekinase-associated protein 1 (SKP1)-Cullin-F-box complex SCF^COI1^ and directs the degradation of JAZs, causing the activation of the JA response. In *Arabidopsis*, there are 12 JAZ members (JAZ1–JAZ12). Under normal conditions, JAZ proteins repress the activity of positive transcriptional regulators by binding to them. Co-repressors are recruited to coordinate the suppression with JAZ proteins. The recruitment of Topless (TPL) is achieved through the novel interactor of JAZ (NINJA), containing an ethylene response factor (ERF)-associated amphiphilic repression (EAR) motif (Figure 2). TPL has been shown to interact with histone deacetylases (HDCAs) to deacetylate histones at promoters or interrupt the Mediator-RNA polymerase II complex, thus leading to the suppression of transcription [118] (Figure 2). There are two major branches of JA signaling downstream of JAZ repressors: the MYC branch, which is responsible for the wounding response, and the ERF branch, which is associated with necrotrophic pathogen resistance (Figure 2). The MYC branch, controlled by MYC-type transcriptional factors, directs wounding responses, including the expression of *Vegetative storage protein 2* (*VSP2*), which is the JA-responsive marker gene. MYC2 recruits MED25, which is one subunit of the plant mediator complex to initiate transcription [119]. The activity of MYC2 could be regulated by the proteasome degradation of JAZs due to the lost and gained repression from JAZs. Furthermore, MYC2 was recently reported to be regulated at the protein level by ubiquitination and deubiquitination [120,121]. MYC2 is the major transcription factor regulating the JA pathway, while MYC3 and MYC4 activate the JA response additively to MYC2, although it is still unclear how MYC3 and MYC4 modulate the JA pathway [122]. MYC2 prefers binding to the G-box sequence (5′-CACGTG-3′). Consistently, the G-box sequence of the promoter is required for the induction of the early-responsive JA response gene *JAZ2*, which is regulated by MYC2. The ERF branch, containing the ethylene response factor 1 (ERF1) and octadecanoid-responsive *Arabidopsis* AP2 59 (ORA59), controls the JA-responsive marker gene plant defensin 1.2 (*PDF1.2*), which is usually regulated by the ET pathway in response to necrotrophic pathogen infection (Figure 2). Interestingly, the MYC branch and ERF branch were found to be antagonistic to each other. The *myc2* mutant was more resistant to the infection of the necrotrophic pathogen *Alternaria brassicicola*, which was probably due to the mediated ERF responses [123]. When the MYC branch was suppressed by *ORA59* overexpression, plants were more susceptible to *P. rapae* larvae, which feed on the plants, causing a wounding response [124]. MYC2 negatively regulates genes in the other JA-responsive branch, including *ERF1*, *ORA59*, and *PDF1.2*. However, the suppression mechanism is still unclear [123]. Interestingly, MYC2 regulates some NAC domain-containing TF genes *ANAC109* and *ANAC055*, which have been shown to positively regulate vegetative storage protein (*VSP*) genes, but negatively regulate *PDF1.2* [125]. Recently, two transcription factors ethylene insensitive 3 and EIN3-like1 (EIL1) upstream of ERF1 and ORA59 were found to be the synergistic convergence knot, which was regulated by both the JA and ET pathways (Figure 2). *EIN3* and *EIL1* are activated by the release of repression from JAZ and HDA6 in the JA pathway; however, they are positively regulated by the essential regulator ethylene insensitive 2 (EIN2) in the ET pathway. EIN2 can stabilize EIN3 and EIL1, via preventing them from being degraded by EIN3-binding F box protein 1 and 2 (EBF1 and EBF2). EBF1 and EBF2 are two F-box proteins mediating the proteasome degradation [126] (Figure 2). Except for the classical COI1-JAZ-MYC2 and COI1-JAZ-EIN3 models, JAZs could bind other transcriptional regulators that are responsive to developmental and environmental cues. JAZs could bind JA-associated MYC2-like 1(JAM1), JAM2, and JAM3. However, unlike MYC2, JAM1, JAM2, and JAM3 negatively regulate the JA pathway through binding to the target sequences of MYC2 as the transcriptional repressors. Probably, these negative regulators were required to fine-tune the overshooting JA response [127,128,129] (Figure 2).

### 3.3. The Ethylene Pathway in Plant Defense

The plant hormone ET is a gaseous hormone, playing key roles in many physiological processes. The biosynthesis of ET originates from S-adenosyl methionine (SAM), which is produced by SAM synthetase from methionine and ATP. 1-Aminocyclopropane-1-carboxylic acid (ACC), the precursor of ET, can be synthesized by ACC oxidase from SAM and converted to ET. It induces the ripening of the fruits or leaf abscission. However, ET also plays an active role in plant defense. For example, in *Arabidopsis*, ET potentiates the expression of *PR1* via an unknown mechanism [130] (Figure 3). There are five ethylene-responsive receptors, which can be divided into two subgroups: the first group contains ethylene response 1 (ETR1) and ethylene response sensor 1 (ERS1), which are characterized by histidine (His) kinase activity; the second group contains ETR2, ERS2, and EIN4, which are characterized by serine (Ser)/threonine (Thr) kinase activity in vitro [131]. These five ethylene receptors are localized on the endoplasmic reticulum (ER) membrane, and they play redundant roles in recognizing ET (Figure 2). A copper ion first associates with the ethylene-binding domain, and is required for high-affinity ethylene-binding activity [132]. In non-challenged cells, these receptors without the association of ET ligands can activate constitutive triple response 1 (CTR1), which negatively regulates EIN2 by phosphorylation [133]. EIN2 can inhibit the degradation of EIN3 and EIL1 by the 26S proteasome, and the degradation process requires EBF1 and EBF2 (Figure 2). EIN3 and EIL1 are positive transcription factors, which regulate ORA59 and ERF1. Upon ethylene binding, CTR1 is deactivated by ethylene receptors such as ETR1, causing a release of the repression on the EIN2, thus stabilizing the EIN3 and EIL1. The JA pathway and ET pathways share the same branch, starting from EIN3 and EIL1. EIN3 and EIL1 can be regulated by both the JA and the ET pathways; therefore, they are synergistic knots of the JA and ET pathways (Figure 2) [126]. The mutant *cev1* of gene constitutive expression of *VSP1 (CEV1)* constitutively activates JA and ET signaling, suggesting a common regulation of both pathways [134]. ET can act on the ERF branch of the JA pathway, but antagonizes the MYC branch. Pathogenesis-related proteins *PR3* and *PR4* are the marker genes of the ET response. *PDF1.2* requires both the JA and ET pathways to be induced, and therefore is a marker gene for both pathways.

### 3.4. SA-JA Cross-Talks

The mutually antagonistic effect between the SA and JA pathways are quite well-known. The suppression of the SA pathway on the JA pathway has been extensively studied [1,77]. NPR1 acts as a crucial modulator in the SA-mediated suppression of JA signaling. It has been reported that the nuclear localization of NPR1 is required for the activation of SA response, but not for the SA-mediated suppression of the JA response [135]. This indicates that cytosolic NPR1 is capable of suppressing the JA pathway. Furthermore, nuclear NPR1 is required for the expression of transcription (co)factors that suppress JA-dependent gene expression such as *glutaredoxin 480* (*GRX480*), *TGA*s, and *WRKY*s [116,136,137] (Figure 3), also suggesting the suppression of the JA response also in a NPR1 nucleus-dependent manner. GRX480 plays a role in SA–JA cross-talk, by suppressing the JA-mediated response, in an NPR1-dependent manner [138]. Additionally, the expression of the marker gene *PDF1.2* in the mutant *npr1*, after the pharmacological application of SA and MeJA, showed that SA suppression on the JA pathway is dependent on NPR1 [139].

Many other regulators have been reported to play roles in the cross-talk of SA and JA pathways, such as the WRKY TFs including WRKY70, WRKY50, WRKY51, WRKY33, and WRKY75, which exert antagonistic influences on SA–JA communication [71,113,114,136,140,141]. Among them, WRKY70, WRKY50, and WRKY51 show suppression of the JA response through an NPR1-independent mechanism [118] (Figure 3), even though they can be regulated by NPR1. Interestingly, ET can bypass NPR1-dependency to render the suppression of SA on JA [139]. Moreover, the exogenous application of SA was reported to suppress the JA pathway through the influence of both *ORA59* expression and protein stability, which targets GCC-box motifs in JA-responsive promoters [137]. However, another recent study indicated that the insects’ egg-triggered SA/JA antagonism is independent from ORA59; however, it relies on the protein stability of MYC2, MYC2, and MYC4 [136].

The influence of the JA pathway on the SA pathway is less clear. The inhibition of the SA pathway by the JA pathway is through the activation of three homologous NAC TF genes: *ANAC019*, *ANAC055,* and *ANAC072* (Figure 3). These can also be activated by the phytotoxin coronatine (COR), which requires SCF^COI1^ and MYC2. These TFs exert an inhibitory effect via the suppression of ICS1, which is a positive regulator of SA synthesis, and they activate the basal transcript level of the SA methyl transferase 1 (BSMT1), which transforms SA into the inactive SA ester (Figure 3). Despite the well-known antagonistic effect, the synergistic actions of the SA and JA pathways have also been frequently reported [135,142,143,144,145,146]. It has been reported that both SA and JA-mediated responses are activated by the epiphytic fungus *Pseudozyma aphidis* [146]. A low light ratio of low red:far-red light compromises both SA and JA-dependent responses in *Arabidopsis*, although the mechanism is still unclear [147]. In the *enhanced disease resistance 1 glucan synthase-like 5 (edr1 gsl5)* double mutant, SA and JA marker genes were synergistically activated, although the single gene mutations *edr1* and *gsl5* only prime the SA response [148]. Exogenous application of low concentrations of both SA and JA can synergistically upregulate both the SA and JA pathways [145].

Reactive oxygen species (ROS), NO, and Mediator, which is a multi-protein complex that functions as a transcriptional co-activator in all of the eukaryotes, are believed to positively regulate both SA and JA pathways. NO was established to have a positive influence on the defense against both biotrophic and necrotrophic pathogens. Ozone exposure to plants generated reactive oxygen species (ROS), and this could trigger all of the SA, JA, and ET responses. ET synthesis preceded both SA and JA production [149]. The loss of function of hemoglobin 1 (*GLB1)*, encoding hemoglobin, led to the enhanced resistance to both biotrophs and necrotrophs, which is concomitant with elevated NO accumulation after both infections [150]. The mutation of Mediator subunit 16 (MED 16) blocked both the induction of JA and SA responses [151]. Furthermore, flagellin (flg22), regulating PTI, can enhance SA, JA, and ET pathways [152,153]. The overall fitness of plants is dramatically impaired, which is concomitant with the activation of the defense responses. The equilibrium of both pathways requires tight control (Figure 3). The regulation of the SA–JA equilibrium is still unclear yet, and it requires more study.

### 3.5. The Involvement of Other Pathways in Plant Defense

In addition to the major roles of the SA, JA, and ET pathways in plant defense, other hormone pathways can mediate plant defense as well. The pathogen entry is facilitated by stomatal opening; in turn, for defense, the plant will close the stomata [154]. It is well-known that the abscisic acid (ABA) pathway can trigger the closure of stomata. Moreover, SA and ABA synergistically mediate the full closure of the stomata, although the synergistic mechanism remains obscure. Furthermore, after wounding or herbivore attack, ABA can act synergistically with the JA pathway, through the positive regulation of the MYC branch of the JA response. Except for its the positive role in defense, ABA was reported to negatively mediate pathogen defense. ABA signaling was found to antagonize the plant immunity by suppressing SA-dependent defenses [8]. Auxin, a major phytohormone operating in plant development, can manipulate plant defense by suppressing SA levels and signaling. In turn, SA signaling can repress the auxin pathway. However, how auxin and SA interact is still unknown. Cytokinins (CKs), which affect plant growth and development, can modulate plant immunity, probably by synergistically influencing the SA response. Gibberellins (GAs) are yet another class of hormones that control plant growth; GA signaling leads to the degradation of growth-repressing DELLA proteins. Interestingly, DELLA proteins have been shown to interact with JAZ proteins to hinder JAZ from inhibiting the activity of transcriptional factors. Thus, GAs negatively influence the JA pathway by the regulation of DELLA degradation [155].

## 4. The Impact of Small Molecules in Plant Defense

The good establishment of high-throughput chemical screening methods allows for the identification of many novel small molecules, protecting plants against infection. The compounds can be generally classified into external and internal compounds according to whether they exist in plants. The external group does not exist in plants. 2,6-Dichloroisonicotinic acid (INA) and acibenzolar-*S*-methyl benzo (1,2,3) thiadiazole-7-carbothioic acid *S*-methyl ester (BTH) are external molecules. The defense genes’ expressions have been shown to be highly induced by the exogenous application of INA and BTH [156]. INA and BTH are thought of as the functional analogues of SA for inducing a similar SA response, especially as they are active in *Arabidopsis* plants overexpressing a salicylate hydroxylase gene (*NahG*), which are unable to accumulate SA. INA and BTH function independently from SA biosynthesis and perception [90,157]. Without further pathogen infection, BABA cannot confer a very obvious impact on SA abundance and defense genes expression in *Arabidopsis*. However, upon infection, pretreatment of BABA can induce stronger SA accumulation and the expression of defense genes such as *PR1* compared to control without BABA treatment [158]. Recently, it was found that IMMUNE-PRIMING CHEMICALS (imprimatins) can block SA glucosyltransferases UDP-glucosyltransferase 74F1 (UGT74F1) and UGT76B1 to control SA accumulation, thus regulating SA-mediated defense responses. [159]. In addition to these external compounds, some internal compounds in plants also play pivotal roles in transducing defense signaling. Amino acid-related isoleucic acid (ILA) was first reported in plants, and decreased after *Pseudomonas* infection [160,161]. Azelaic acid (AzA) with the structure of nine-carbon dicarboxylic acid accumulates in petiole exudates in *Arabidopsis* leaves upon infection. Furthermore, AzA can protect plants against infection in both local and systemic leaves. These results strongly suggest a key role of AzA in the defense response [162]. Pipcolic acid (Pip) is a non-proteinaceous product of lysine catabolism that was increased to high levels in pathogen-inoculated leaves. With a compromised defense immunity in the mutant *ald1*, the loss of function of aminotransferase aberrant growth and death 2 (AGD2)-like defense response protein 1 (ALD1) could be rescued by the application of Pip. This indicates that ALD1 might be involved in the synthesis of Pip [163,164]. Indeed, L-lysine lost an α-amino group to become 2,3-de-hydropipecolic acid (2,3-DP) through transamination by ALD1. Then 2,3-DP is then reduced to Pip by SAR-deficient 4 (SARD4) and an unknown reductase [165,166]. However, Pip-triggered defense is dependent on Flavin-dependent mono-oxygenase 1 (FMO1), since the exogenous application of Pip could not activate defense in the FMO1 mutant. Recently, *N*-hydroxypipecolic acid was found to be the product of FMO1 from Pip, and it can accumulate upon infection. Furthermore, it can restore the resistance against infection for the FMO1 mutant. Thus, a novel internal molecule, *N*-hydroxypipecolic acid, was revealed to regulate defense, suggesting a close connection between amino acids and defense [167,168].

## 5. SAR and Priming

### 5.1. Systemic-Acquired Resistance

Systemic acquired resistance (SAR) is a broad-spectrum resistance that occurs at systemically in leaves following ETI. Due to the activation of SAR, plants will stay alarmed and be prepared for further infections for a time over weeks to months. Some signals are obviously passed from locally-infected tissues to systemic leaves. The initiation of SAR in systemic leaves is accompanied by the massive transcriptional programming of SA-related genes, such as *PR1*, *PR2*, and *PR5*. Although SA is required for SAR, it is not a mobile signal. Instead, other signals such as methyl salicylic acid (MeSA), AzA, glycerol-3-phosphate (G3P) and dehydroabietinal (DA), an abietane diterpenoid, are mobile, and they could induce SA accumulation and trigger the expression of pathogen-related genes in systemic leaves [14]. FMO1 is required for SA accumulation in systemic leaves and the triggering of SAR; however, it is not required for SA synthesis in the local leaves. An exciting finding showed that a non-protein amino acid, Pip, was catalyzed by FMO1 to become *N*-hydroxy-pipecolic acid, indicating the role of Pip and its product in transducing SAR through FMO1 [167,168,169,170]. Interestingly, the mutant of *Arabidopsis* mediator subunit 16 (MED16) was compromised in SAR, and it probably acts as the bridge between NPR1 and RNA polymerase II transcription machinery [151].

### 5.2. Defense Priming

Defense priming is a status in plants without the obvious induction of defense; however, it can activate defense mechanisms more rapidly and robustly upon the pathogen infection. The intensity of the defense response is usually at the cost of plant health. The resistance effect in primed plants can be comparable as in constitutively defense-elevated plants; however, it does not have a fitness cost. Thus, defense priming has a more practical use in agriculture without influence on yields. Defense priming is believed to be relevant to ROS production, transcription programming, and modification on an epigenetic level, which can facilitate the switch-on defense response very rapidly upon infection [171]. For instance, BABA can prime SA accumulation and the expression of defense genes such as PR1 [158]. BABA-induced resistance is a slightly complicated. The protection by BABA against *Pseudomonas* is somehow compromised in *salicylate hydroxylase gene* (*NahG)* and *npr1*, suggesting a dependence on the SA pathway. BABA-induced priming was shown to be partially dependent on abscisic acid (ABA) in *Arabidopsis*, since BABA-primed resistance was impaired in *aba1-5* and *abi4-1* mutants, which are deficient in ABA signaling. In addition, in the *gsl5* mutant of a callose synthase gene glucan synthase-like 5 (the same as powdery mildew resistant 4), BABA-induced callose accumulation and resistance to *Alternaria brassicicola* and *Plectosphaerella cucumerina* is blocked [172]. Additionally, BABA-induced resistance against Pseudomonas was found via specific antagonism on coronatine, but not JA [173]. BABA was recently shown to prime for the induction of pattern-triggered immunity marker genes as well [36]. It is surprising that the BABA-primed-defense state can be inherited to the next generation [174]. Interestingly, in the case of the above-mentioned SAR, some genes are not directly induced in systemic leaves; however, they are primed and can be augmented upon pathogen challenge, such as phenylalanine ammonia lyases (PALs) in *Arabidopsis*. In *Arabidopsis*, priming for potentiated PALs expression could be mimicked by pretreatment with low dosages of the synthetic SAR inducer BTH, which resulted in the augmented accumulation of messenger RNA (mRNA) of PALs after infection with virulent *Pseudomonas syringae pv.tomato* DC 3000 [175]. Hexanoic acid was found to prime the defense in *Arabidopsis,* in order to protect against *B.cinerea* [176]. The primed defense response can be seen in the mutant of the gene enhanced disease resistance 1 (EDR1). The *edr1* mutant led to the enhanced resistance to *Pseudomonas syringae* and *powdery mildew*, suggesting a negative role in plant defense. However, unchallenged by pathogens, *edr1* did not show significantly elevated SA levels and *PR1* expression, instead resembling the primed state in BABA-treated plants [98].

## 6. Questions Remaining and Perspectives

Researches in *Arabidopsis* has significantly elaborated upon its different defense mechanisms. In this review, different layers of defense mechanisms are revealed. The classical hormone pathways against defense are intensely discussed. Extensive studies have demonstrated that these different layers of plant defense are linked to hormone pathways, especially for the SA and JA/ET pathways. However, the connections between the hormone pathways and other above-mentioned defense aspects will need further investigation. For physical defense, the stomata rapidly close less than one hour after infection, in order to prevent pathogen entry [177]. The hormones SA and ABA additively play positive roles in closing the stomata; however, the JA pathway tends to switch on its stomata [178,179,180,181]. The stomata respond to pathogen invasion quickly, leading to the hypothesis that some signals might exist in the guard cells. These signals might be the hormones themselves, or other molecules. Therefore, real-time measurements of hormones in guard cells, especially during early infection, will be pivotal. However, single cell measurements of hormones will also require new techniques.

We also need to know how hormone pathways participate in plant immunity. The bridges of how PRRs mediate hormone pathways are still missing, although some connections are revealed for the SA pathway. For instance, some PRRs, such as pattern-triggered immunity compromised receptor-like cytoplasmic kinase 1 (PCRK1) and PCRK2, are required for SA biosynthesis upon infection [182]. However, how PCRK1 and PCRK2 can mediate SA biosynthesis is not fully understood. Moreover, hormone pathways, including the SA, JA, and ET pathways, are required for full PTI and ETI against *Pseudomonas syringae* [183]. This was elaborated by the loss of resistance only in the quadruple mutant, which is simultaneously deficient in SA, JA, ET, and phytoalexin 4 (PAD4); however, this is not the case in the single mutants [183,184]. This suggests that these pathways play redundant roles in PTI and ETI; however, how they interact with each other to mediate defense is still unknown. More efforts can be taken, especially for how PRR-mediated defenses pass signals to induce these hormone pathways. Moreover, the timing and spatial mapping of different hormone accumulations and the interactions among SA, JA, and ABA pathways during PTI and ETI should be investigated more closely.

During the co-evolution of pathogens and plant hosts, hormone pathways cannot only protect against pathogens invasion, they can also be utilized by pathogens to facilitate disease. Coronatine (COR), which is produced by *Pseudomonas syringae*, is the functional mimic of JA-Ile, and therefore induces the JA pathway to suppress the SA pathway for bacterial infection [185]. Other COR-like molecules are produced in *Streptomyces scabies* and *Xanthomonas campestris* pv. *Phormiicola* [185,186]. How can plants cope with these compounds to fight back? AvrPtoB was previously shown to degrade the flg22 receptor, flagellin-sensitive 2 (FLS2) [65]. Recently, a study showed that the master player NPR1 in the SA pathway is targeted for degradation by AvrPtoB [187]. This suggests that AvrPtoB redundantly interferes with different layers of defense, including PAMP recognition and the SA pathway, to suppress plant immunity. However, in turn, plants may develop a strategy to overcome the new strategy from AvrPtoB, which requires further study. NPR3/NPR4 are also SA receptors, and they are new key players in the SA pathway [95]. It is possible that NPR3 and NPR4 are interfered with by virulent effectors. Furthermore, other key players in the JA pathway and ET pathway, such as MYC2 and/or EIN2, might be attacked by virulent effectors. Thus, more strategies for pathogens to trick host plants with in hormone pathways require further studies to better help in the improvement of plant resistance.

Other defense mechanisms, such as SAR and priming, might be manipulated to improve plant defense without the loss of yield [14]. Recently, amino acid-related small molecules were intensively explored, and found to be closely related with plant defenses such as ILA, Pip, and its product, *N*-hydroxypipecolic acid [165,167]. SAR, priming, and Pip-regulated defenses have been revealed to at least partially rely on the SA pathway [164,171,188]. The ABA pathway is required, for instance, for BABA-induced priming [189]. The roles of hormone pathways in these defense phenomena, and their detailed mechanisms, require further studies.

## Figures and Tables

**Figure 1 cells-07-00252-f001:**
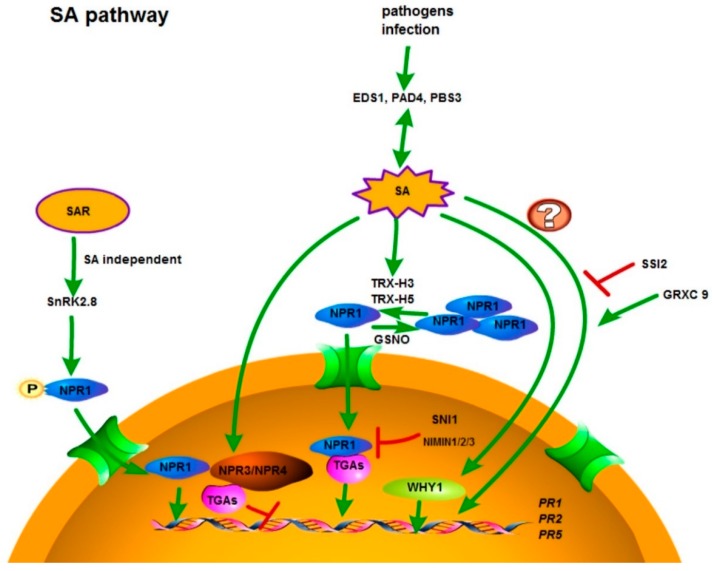
Model of salicylic acid (SA) synthesis and SA signal transduction. Enhanced disease susceptibility 1 (EDS1), phytoalexin-deficient 4 (PAD4), and avrpphb susceptible 3 (PBS3) are involved upstream of SA synthesis, and can be induced by positive SA feedback. The redox state alteration caused by the SA burst leads to the monomerization of non-expressor of PR genes 1 (NPR1). The monomerization of NPR1 by thioredoxins (TRXs) such as TRX-H3 and TRX-H5 cause the translocation of NPR1 in the nucleus and subsequent interaction with the positive transcription activators TGACG sequence-specific binding proteins (TGAs), the negative regulators non-inducible immunity (NIM1)-interacting proteins (NIMINs) and Suppressor of *npr1*-inducible 1 (SNI1). NPR1 is imported into the nucleus through nuclear pores. S-nitrosoglutathione (GSNO) can facilitate the oligomerization of NPR1. During systemic acquired resistance (SAR), the kinase SnRK2.8 is required for the phosphorylation of NPR1, which is not triggered by SA; it is triggered by an unknown signal. A possible working model would be that SA triggers the formation of NPR1 monomers. Then, NPR1 monomers are phosphorylated by sucrose non-fermenting 1 (SNF1)-related protein kinase 2.8 (SnRK2.8) for its nuclear import. The transcription factor whirly 1 (WHY1) is probably involved in the NPR1-independent SA response. Suppressor of SA insensitivity 2 (SSI2) is a negative regulator, independent from NPR1, which regulates the SA-mediated response. Overexpression of glutaredoxin C9 (*GRXC9*) could regulate the SA pathway independent from NPR1. Pathogen-related 1 (*PR1*), *PR2,* and *PR5* are SA-responsive marker genes. Activation (closed arrowhead), suppression (⊥), and important genes are indicated.

**Figure 2 cells-07-00252-f002:**
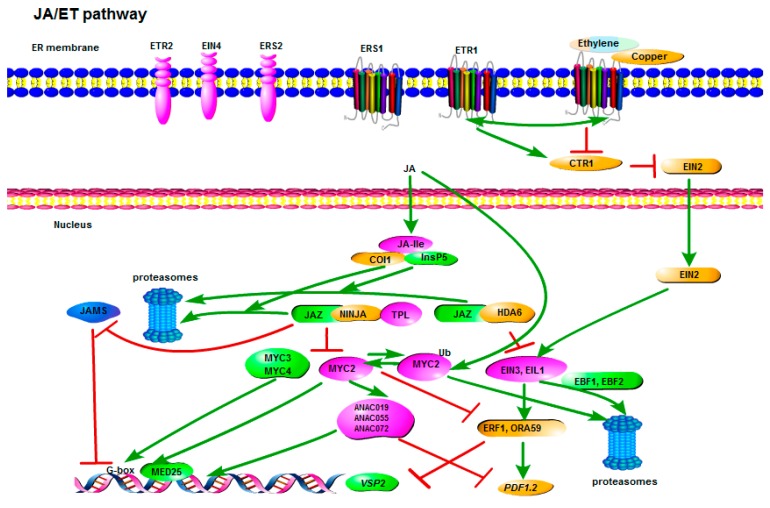
Model of biosynthesis and signal transduction of jasmonic acid (JA) and ethylene (ET). JA and ET can synergistically activate ethylene insensitive 3 (EIN3) and ethylene insensitive 1 (EIL1), which are positive regulators of ERF1 and octadecanoid-responsive *Arabidopsis* AP2 59 (ORA59), leading to the induction of plant defensin 1.2 (*PDF1.2*). The active form jasmonoyl-l-isoleucine (JA-Ile) can be recognized by coronatine insensitive 1 (COI1) and the co-receptor Inositolpentakis phosphate 5 (InsP5), causing the degradation of jasmonate ZIMs (JAZs). There are two antagonistic branches in the JA response: the MYC branch and the ERF branch. JAZ proteins can suppress the activity of MYC2, the positive regulator of *Vegetative storage protein 2* (VSP2), by enrolling the negative regulator Topless (TPL) via the novel interactor of JAZ (NINJA). MYC2 can directly interact with mediator 25 (MED25, which is a subunit of the mediator. MYC2 can activate the expression of NAC domain containing proteins (ANACs) to upregulate *VSP2*. MYC3 and MYC4 can activate JA response additively with MYC2 by binding to the G-box. JAZs could also suppress the activity of jasmonate associated MYC2 like proteins (JAMs), which are a group of transcriptional repressors that could bind to the MYC2 target sequence G-box to negatively regulate the JA response. The MYC2 protein is regulated at protein levels. MYC2 can be ubiquitinated during the JA response for degradation. Thus, JA-mediated defense is suppressed. Conversely, MYC2 can be inversely deubiquitinated to stay stable, triggering a positive regulation on JA response. In the ERF branch, JAZs can inhibit the activity of EIN3 and EIN3-like 1 (EIL1) by direct association with them and enrollment of the histone deacetylase 6 (HDA6). In the ET pathway, there are five ethylene receptors localized in the ER membrane. A copper ion interacts with the ethylene-binding domain. The interaction is required for high-affinity ethylene-binding activity. By binding to ET, constitutive triple response 1 (CTR1) is deactivated by ET receptors. CTR1 can inhibit EIN2, which activates EIN3 and EIL1 by preventing their degradation of them via EIN3-binding F box protein 1 and 2 (EBF1 and EBF2). Activation (closed arrowhead), suppression (⊥), and important genes are indicated.

**Figure 3 cells-07-00252-f003:**
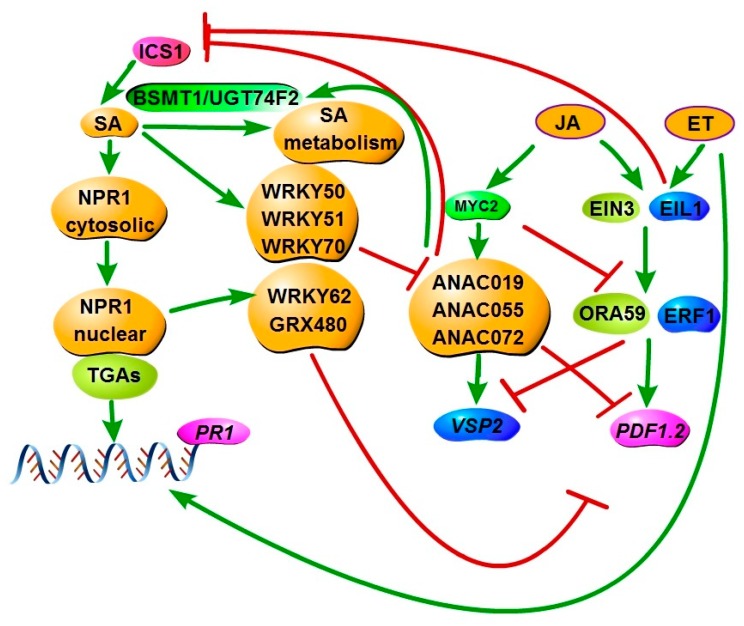
Interactions between the SA and the JA/ET pathways. ET can synergistically induce *PR1* via an unknown mechanism. SA and JA/ET are mostly antagonistic to each other. On the left-hand (SA) side, the cytosolic localization of NPR1 is sufficient for the suppression of the JA/ET response. Some components that are regulated by NPR1, such as WRKY62, TGAs, and *glutaredoxin 480* (GRX480), can confer the suppression on the JA response. Some WRKYs such as WRKY50, WRKY51, and WRKY70, can suppress the JA response independent from NPR1. The suppression of the JA pathway by the SA pathway is downstream of JA synthesis, having no influence on JAZs accumulation; however, it results in the degradation of ORA59. On the right-hand (JA) side, JA can suppress the SA response by inducing *ANAC*s, which are regulated by *MYC2*. ANACs can inhibit isochorismate synthase 1 (ICS1); however, they can enhance the basal transcript level of the SA methyl transferase 1 (BSMT1) and UDP-glucosyltransferase 74F2 (UGT74F2) to elevate SA accumulation. Furthermore, EIN3 and EIL1 can suppress ICS1 activity by directly binding to the promoter of *ICS1*. Activation (closed arrowhead), suppression (⊥), and important genes are indicated.

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
