# Peer review of "Different Pathogen Defense Strategies in Arabidopsis: More than Pathogen Recognition"

_cells, 2018, doi:10.3390/cells7120252_

Round 1
Reviewer 1 Report
The manuscript was not well written, including disordered, wrong demonstration, old references, missed references and knowledge points. Most importantly, this manuscript lacks an in-depth synthesis of the literature including critical analysis; challenges and questions still to be addressed, future direction discussion.
My major concerns are as following:
1. Old references: The purpose of a review paper is to review new progress in a particular topic. Most of references were published before 2016. Many new advances were missed.
For example:
Line 103-104, the authors missed the new progress on RLPs research (please see reference See: J Exp Bot. 2016 May; 67(11):3339-51. doi: 10.1093/jxb/erw152.).
Part 1.3 Virulent effector induced plant defense: too old references about the research on effectors of AvrPtoB and HopM1.
2. Reference missed: For many demonstrations, the authors need to give the references.
3. Cite too many review papers: The emphasis of a review paper is interpreting the primary/original literature on the subject. The authors should demonstrate the progress and points by citing original papers, not directly from the review paper,
such as line 88-91; Line 110-115; Line 157-160
4. Wrong demonstration:
Line 75-77: Chitin, produced by fungal cell wall is detected by chitin elicitor receptor kinase 1 (CERK1), another receptor like kinase to regulate pathogen defense, although the downstream signaling is still unclear? This is wrong demonstration. See Plant Cell Physiol. 2017 Jun 1;58(6):993-1002. doi: 10.1093/pcp/pcx042 and Plant Signal Behav. 2017 Sep 2;12(9):e1361076. doi: 10.1080/15592324.2017.1361076.
Line 108: how these virulent effectors suppress PTI is still unclear? Some effectors have been studied clearly how to suppress PTI.
5. A lot of knowledge points were missed:
For example:
Physical defense (stomata and lenticels);
Chemical defense (plant defensins, Phytoanticipins)
PRRs (FLS3,XA21, Ve1, ….. )
6. Most importantly, this manuscript lacks an in-depth synthesis of the literature including critical analysis; challenges and questions still to be addressed, future direction discussion. In my opinion, this manuscript mostly like science summary.
Minors:
Line 20: PAMP (pathogen associated molecular pattern)
Line 37: Erysiphe Orontii and Erysiphe Cichoracearum (Referene?)
Line 40”: which kill hosts using toxic metabolites, enzymes or microRNAs (Need to add more reference to cover these demonstrations).
Suggest the authors to add one paragraph to demonstrate PTI, ETI and the classical Zigzag Model following 1. Different layers of plant defense.
Line 52: consisting of lipid, hydrocarbon polymers and wax (Reference?).
Line 59: Phytoalexins are broad-spectrum inhibitors, acting as antimicrobial toxins (reference?).
Line 66: pathogen-associated molecular pattern
Line 66: (PRRs)
Line 70 and 77: EF-Tu
Line 82: However, most of the PAMP/PRR pairs are unclear. Which PAMPs? Which PRRs?
Line 79-91: disordered description.
Line 84:FLG22-induced receptor-like kinase 1 (FRK1)
Line 94: ET?
Line 103: involvement of defense in tomato and Arabidopsis (Cite original papers?).
Line 104: Fifty-six putative RLPs were reported to exist in Arabidopsis?
Line 107: mechanism that
Line 167-168: abrupt description: These pathogen-related (PR) genes such as PR1 encode many antimicrobial proteins.
Author Response
The pont-to-point response is as below.

Reviewer 2 Report
The review paper entitled "An overview of different aspects of pathogen defense: PTI, ETI, hormone pathways, SAR 2 and priming" by Zhang et al. describes and summarised some of molecular aspects associated with plant defense responses upon pathogen attack. I think this review is well-written, stuctured and covered of all basic moments with plant signaling system. I do not have any principal comments on this manuscript and hope that it will be most readable and useful for researchers working in Plant Sciences. A single comment I have: the authors cited 134 references, but there are only less than a decade has been published last 2 years. I beleive that "Molecular plant signaling" is a quickly developing direction in science, thereby annual quantity of papers that are devoted to molecular mechanisms of plant defense to biotic stresses can be achieved up to 100 and more (according to Medline database). The authors should revise ther paper to include more actual articles published last 3 years.
Author Response
The review paper entitled "An overview of different aspects of pathogen defense: PTI, ETI, hormone pathways, SAR 2 and priming" by Zhang et al. describes and summarised some of molecular aspects associated with plant defense responses upon pathogen attack. I think this review is well-written, stuctured and covered of all basic moments with plant signaling system. I do not have any principal comments on this manuscript and hope that it will be most readable and useful for researchers working in Plant Sciences. A single comment I have: the authors cited 134 references, but there are only less than a decade has been published last 2 years. I beleive that "Molecular plant signaling" is a quickly developing direction in science, thereby annual quantity of papers that are devoted to molecular mechanisms of plant defense to biotic stresses can be achieved up to 100 and more (according to Medline database). The authors should revise ther paper to include more actual articles published last 3 years.
àYes, we have totally agreed on the comment that we do not have enough new references. Thus, we have added around 60 more references to support our paper. We have cited 40 more papers, which were published after 2016.
Reviewer 3 Report
Please, check my comments above

Author Response
Point to point response are as below.

Round 2
Reviewer 1 Report
Addressed my concerned
Reviewer 3 Report
Dear authors,
I enjoyed reading the new version of the manuscript, which has substantially improved from the previous version I have been reading and correcting.
Now, the language is coherent with the canonical scientific writing and the references have been added and updated.
Therefore, i have no further comments to add.